# Context-Aware Multi-Agent Safety Evaluation for Autonomous Driving

## Abstract

Autonomous Driving (AD) faces persistent safety challenges from unforeseen long-tailed driving scenarios that require massive evaluation. Existing solutions, such as road test, scenario-based simulation and rule-based verification, remain insufficient: they either fail to uncover hazardous edge cases and inherit unsafe habits from human data, or lack adaptability across regions. Additionally, current approaches often provide limited contextual understanding, making it challenging to generate interpretable explanations of unsafe behavior. To address these gaps, we introduce **DriveEval**, a context-aware multi-agent framework for autonomous driving safety evaluation. It leverages the comprehensive knowledge and reasoning ability of large language models (LLMs) to understand traffic scenes and detect edge cases, while applying context engineering to ground LLMs in external knowledge, including traffic rules and historical accident data, for interpreting unsafe driving behaviors. The framework is organized as a multi-agent workflow comprising a Data Annotator, Scene Extractor, Rule Checker, Accident Retriever, and Driving Assessor, each handling specialized functions. This multi-agent design improves precision through specialization, enables modular expansion with new knowledge sources, and allows the most suitable model to be chosen for each task, offering stronger performance than a single monolithic agent. Experiments show that DriveEval can evaluate sensor data, such as dashcam video, to identify safety risks and recommend actionable improvements. Its assessments are closely aligned with human annotations, demonstrating that context-aware evaluation provides interpretable safety assurance.

## 1 Introduction

Autonomous Driving (AD) is poised to redefine transportation, with profound implications for enhancing road safety, traffic efficiency, and personal mobility for a diverse population, including the elderly and disabled. The central promise of AD is the mitigation of human error, which accounts for over 90% of traffic accidents. (Chougule et al., 2024) However, the path to full, unconstrained autonomy (SAE Levels 4 and 5) is impeded by the arduous challenge of ensuring safety, particularly when faced with unforeseen "long-tail" driving scenarios. (Wang et al., 2020; Liu & Feng, 2024) Even mature systems exhibit vulnerabilities; for instance, Waymo vehicles have incorrectly interpreted a pickup truck being towed at an angle, leading to two separate collisions within minutes (Waymo, 2024). Tesla's Full Self-Driving (FSD) software has demonstrated hazardous behavior at railroad crossings, including failing to stop during arm descending and light flashing Ingram, 2025. Furthermore, as end-to-end AD models learn from vast amounts of data collected from their fleet, they risk imitating unsafe human driving priors, such as aggressive lane changes or tailgating. These persistent safety concerns hinder widespread deployment and underscore the critical necessity for a more sophisticated and rigorous safety evaluation paradigm.

Existing paradigms for AD safety evaluation approaches, however, are insufficient to provide the requisite level of assurance. On-road testing is the most direct and realistic method of evaluation, but demonstrating reliability superior to human drivers would require a fleet to traverse hundreds of millions to billions of miles, a time- and resource-prohibitive endeavor. (Kalra & Paddock, 2016) Moreover, such vast mileage does not guarantee the observation of rare yet critical hazardous events. Simulation-based testing offers a scalable and cost-effective alternative for exploring dangerous edge cases, but its efficacy is often compromised by the "sim-to-real" gap. This means discrepancies in

environmental, sensor, and physics models fail to capture real-world complexities, thus its scope is limited to pre-defined parametric variations. (Kaur et al., 2021) Formal methods aim to provide mathematical guarantees by verifying system properties against a specification, ensuring that undesirable events like collisions cannot occur under a given set of assumptions. Despite their rigor, the practical application of these methods is constrained by challenges in accurately modeling the non-determinism of the real world, the brittleness of assumptions, and state-space explosion issues that limit scalability. (Mehdipour et al., 2023) In addition, a fundamental gap shared by these approaches is their lack of deep contextual understanding and interpretability. They typically yield numerical metrics, such as time to collision and crash rates, but fail to provide interpretable explanations for why an AD behavior is deemed unsafe.

The limitations of current methodologies reveal a pressing need for a new evaluation paradigm founded upon several key principles. P1) Such a framework must be data-effective, capable of proactively identifying latent risks from daily driving scenarios, as collisions and critical safety events are exceedingly rare in raw data. P2) It should be grounded in realism, leveraging real-world driving data to circumvent the sim-to-real gap and uncover unexpected, emergent risks. P3) The system must possess strong generalization and reasoning abilities to interpret the complex, unpredictable dynamics of traffic, including the nuanced interactions and intentions of various road users. Besides, out-of-distribution edge cases should be robustly handled. P4) Furthermore, its assessments should be grounded in well-founded, verifiable knowledge bases, such as statutory traffic regulations and empirical data from historical accidents, to ensure consistent and authoritative criteria. P5) This knowledge should be adaptable to local traffic laws and regional driving norms. P6) Finally, the framework should produce explainable evaluation results, moving beyond opaque metrics to offer transparent, interpretable feedback.

To address these needs, we propose **DriveEval**, a context-aware multi-agent framework for autonomous driving safety evaluation. DriveEval evaluates driving safety from sensor data, such as dashcam video, GPS, and vehicle telemetry, captured during daily operation, thereby ensuring its analysis is grounded in real-world conditions. [P2] The framework orchestrates a sequence of specialized agents: The Data Annotator leverages the strong zero-shot generalization capabilities of Vision Language Models (VLMs) to process raw sensor data into rich, multi-faceted traffic descriptions, capturing participant interactions, environmental conditions, and rare events. [P3] The Scene Extractor then decomposes these complex narratives into structured, non-overlapping scenes to streamline the analysis for downstream agents. The Rule Checker employs an agentic Retrieval-Augmented Generation (RAG) pipeline to enforce legal context, querying a vector database of regional traffic regulations to ensure compliance and facilitate adaptation to local rules. [P4, P5] Concurrently, the Accident Retriever identifies analogous risks by querying a historical accident knowledge graph using graph RAG, excelling at matching the structured representation of the current traffic scene to semantically similar scenarios in the accident database. Thus, it enables the proactive identification of latent hazards even in the absence of an immediate safety-critical event. [P1, P4] Finally, the Driving Assessor leverages the advanced reasoning capacity of LLMs to synthesize these diverse analytical perspectives into a holistic, interpretable safety report, detailing the system's performance, identifying strengths and weaknesses, and providing actionable advice for improvement, in stark contrast to opaque numerical metrics. [P3, P6]

The main contributions of this paper are:

- A novel context-engineered, multi-agent framework, **DriveEval**, for interpretable AD safety evaluation. Its modular architecture, comprising specialized agents for data annotation, scene extraction, rule checking, and accident retrieval, allows individual components to be optimized with the most suitable models and adapted to new contexts.

- A data annotation workflow and a manually annotated dataset for evaluating context-aware safety frameworks. We developed a semi-automated tool to generate draft annotations for human refinement, providing a valuable resource for benchmarking and enriching the historical accident knowledge base. We will release our dataset to facilitate reproducible research.

- A systematic empirical analysis of the framework's agents, evaluating various LLMs to determine the optimal configuration for overall performance and providing insights into the design of effective multi-agent evaluation systems.

- Case studies of the proposed framework demonstrates its alignment with the core principles required for future AD safety evaluation. We show that its outputs are highly explainable and action-

able, providing a practical tool for developers to diagnose systemic weaknesses and guide targeted improvements.

The remainder of this paper is organized as follows. Section 2 reviews related work in detail. Section 3 describes the architecture and components of the DriveEval framework. Section 4 presents our experimental setup, dataset, and results. Finally, Section 5 discusses the implications of our work and concludes the paper.

## 2 RELATED WORK

**Autonomous Driving Paradigms and Safety Challenges** Autonomous driving (AD) systems follow two main paradigms: modular pipelines and end-to-end models. Modular stacks decompose perception, planning, and control, offering interpretability and component-level verification, but suffer from error propagation across modules. End-to-end models map raw inputs directly to controls, reducing engineering overhead but remaining opaque "black boxes" prone to brittle failures on out-of-distribution long-tail events. (Wang et al., 2020; Liu & Feng, 2024; Zhao et al., 2025) Across paradigms, AD systems face three key challenges. (*i*) Data-driven models suffer from pathologies such as *unsafe-habit overfitting*, where policies replicate unsafe behaviors from human demonstrations, and *edge-case underfitting*, where rare but hazardous scenarios are poorly generalized. (Fu et al., 2024) (*ii*) Safe interaction with human road users remains an unsolved challenge, as their behaviors are diverse, rapidly evolving, and often irrational. (Wang et al., 2022) (*iii*) Interpretability is lacking: both hierarchical stacks and end-to-end models struggle to provide causal explanations of failures, limiting debugging, regulatory oversight, and public trust. (Teng et al., 2022)

**Existing AD Safety Evaluation Methods and Metric Limitation** Researchers employ diverse validation methods to ensure safety, each with limitations. On-road testing is most faithful but infeasible at scale. Track testing enables repeatable hazards with virtual actors but remains costly and narrow. Simulation dominates for scalability and safety, supporting both common and rare scenarios via fuzzing and adversarial generation (Ren et al., 2025), yet the sim-to-real gap persists (Ding et al., 2023). Formal methods offer guarantees but scale poorly to perception-heavy systems, while accelerated statistical techniques reduce mileage at the cost of distributional assumptions. Safety is usually measured by perception accuracy, trajectory safety, risk indicators (e.g., Time-to-Collision), law compliance, or disengagement rates. These metrics are surface-level: a low Time-to-Collision may stem from perception errors, flawed planning, or defensive driving, but cannot be distinguished. Such non-interpretability limits their actionability (Sharath & Mehran, 2021). Hence, moving toward interpretable, evidence-backed explanations is key to linking evaluation with retraining and system design (Lai et al., 2025).

**Enabling Technologies for a New Paradigm** Recent advances in AI enable more contextual and interpretable evaluation of AD safety. LLMs and VLMs now demonstrate strong reasoning and grounding, allowing them to annotate scenes, infer intentions, and generalize to rare events, offering richer insights than traditional perception metrics. Their zero-shot generalization further covers the long-tail of scenarios without exhaustive retraining (Tian et al., 2024; Cao et al., 2024). Multi-Agent Systems (MAS) naturally decompose evaluation into specialized roles—e.g., rule-checking, accident retrieval, scene annotation—while a central agent integrates outputs. This modular design improves robustness, scalability, and interpretability compared to monolithic evaluators (Wu et al., 2025). Finally, context engineering with RAG grounds assessments in external knowledge such as traffic codes, regional regulations, and accident databases, ensuring provenance and adaptability (Mei et al., 2025; Yuan et al., 2024; Hussien et al., 2025). Beyond transparency, it produces diagnostic explanations (e.g., citing a violated law or similar past collision) that directly guide retraining and policy adjustment, turning evaluation into an active driver of system improvement.

## 3 METHODOLOGY

### 3.1 PROBLEM FORMULATION

We formulate the task of context-aware safety evaluation as learning a mapping from raw driving data to a structured, interpretable safety report, grounded in external knowledge. Formally, let a driving log $L$ represent a continuous driving session, composed of a time-series of multi-modal sen-

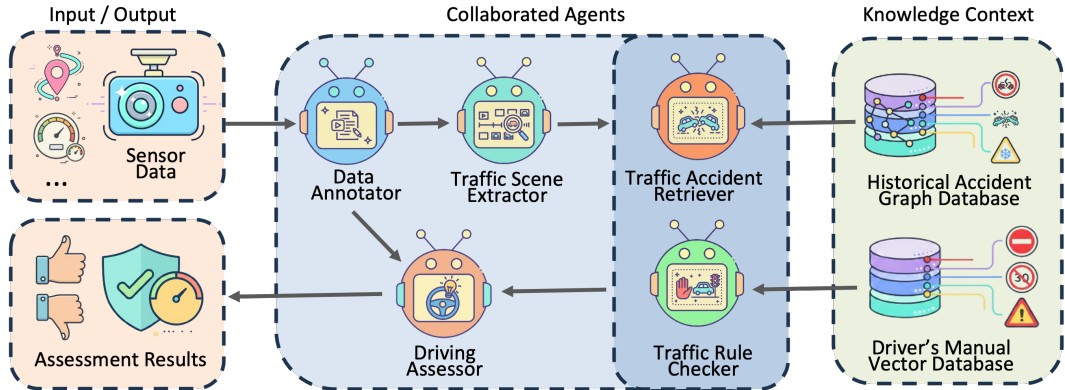

Figure 1: The multi-agent workflow of the DriveEval framework. Sensor data is processed through a pipeline of collaborating agents that query external knowledge bases to produce a comprehensive safety assessment.

sor data. This primarily includes a video stream $V = \{v_1, v_2, \cdots, v_T\}$ and may be augmented with corresponding telemetry data $M = \{m_1, m_2, \cdots, m_T\}$, such as GPS and vehicle speed. The evaluation system has access to two external, structured knowledge bases: a corpus of traffic regulations, $K_R$, specific to a geographic location, and a database of historical accident and near-miss scenarios, $K_A$.

The desired output is a structured and interpretable safety report, $R$, defined as a set of evaluation results $\{r_1, r_2, \cdots, r_n\}$. Each result contains: (1) A quantitative safety score and a categorical risk level summarizing the overall safety performance. (2) A holistic narrative evaluation that synthesizes key events and provides an overarching assessment. (3) A distinct list of identified strengths, corresponding to positive and safe driving behaviors observed. (4) A distinct list of identified weaknesses, corresponding to unsafe behaviors or errors, where each weakness is ideally supported by evidence from the knowledge bases $K_R$ or $K_A$. (5) A set of actionable improvement advice designed to directly address the identified weaknesses.

The core objective is to build a mapping function, $F(L, K_R, K_A) \to R$, that transforms raw driving data into this rich, multi-faceted report. The function must not only detect salient safety events but also reason about their context to generate a synthesized assessment that is quantitative, qualitative, and directly actionable.

### 3.2 FRAMEWORK OVERVIEW AND WORKFLOW

To solve the formulated problem, we propose **DriveEval**, a context-aware multi-agent framework that emulates a "driving analyst" by integrating the reasoning capabilities of Large Language Models with specialized domain knowledge. The framework, shown in figure 1, is predicted on a modular, agent-based design philosophy where a pipeline of coordinated agents breaks down the complex evaluation task into manageable subtasks. This design enhances interpretability and allows for the optimization of each component with the most suitable model. Crucially, the framework leverages context engineering to ground its two forms of external knowledge: (1) Normative Knowledge (traffic regulations) and (2) Experiential Knowledge (historical accidents).

The framework operates as an end-to-end pipeline that transforms raw driving data into a final evaluation report. The workflow proceeds as follows: The system takes a continuous driving log $L$ as input, primarily dashcam video $V$ and corresponding telemetry data $M$ if any. This continuous video is automatically segmented into a sequence of shorter, manageable clips (e.g., 10 seconds each) for focused analysis. Each clip is processed by a sequence of agents. The **Data Annotator** first generates a detailed textual description of the video clip. This annotation is then analyzed concurrently by the **Traffic Rule Checker** and the **Traffic Accident Retriever** to identify normative violations and experiential risks. The findings for each clip are passed to the **Driving Assessor**, which generates suggestions and a structured assessment. After all clips are processed, a supervisor agent aggregates these individual assessments into the final, comprehensive report $R$. The final report is designed

to be actionable, providing a clear "audit trail" of weaknesses that can feed back into the development cycle. These insights can be used for human-in-the-loop debugging or to automatically curate targeted scenarios for retraining.

### 3.3  AGENT AND COMPONENT DETAILS

#### 3.3.1  DATA ANNOTATOR: VLM-BASED TRAFFIC VIDEO UNDERSTANDING

The first and most critical agent in the DriveEval pipeline is the Data Annotator. Its primary function is to transform raw, unstructured visual data from dashcam video clips into a rich, structured textual representation. This process converts pixel-level information into a high-level semantic summary, providing the foundational context upon which all subsequent reasoning and evaluation agents depend.

The core of the Data Annotator is a meticulously engineered prompt designed for a powerful Vision-Language Model (VLM). The development of this prompt followed a two-stage methodology to ensure both comprehensive coverage and alignment with our evaluation principles. First, an LLM was tasked with summarizing authoritative sources, including various driver's manuals and safety reports from the National Highway Traffic Safety Administration (NHTSA), to automatically extract a taxonomy of factors that critically affect driving safety. This data-driven approach produced a draft prompt grounded in established safety knowledge. Second, this draft was manually refined to explicitly target the key principles of our evaluation paradigm, ensuring the VLM's output would be data-effective, realistic, and contain the necessary information for explainable, knowledge-grounded assessments. The final prompt directs the VLM to analyze each video clip from four distinct, complementary perspectives, shown as figure 2: (i) the state of traffic signals and signs; (ii) the dynamic interactions and inferred intentions of nearby participants; (iii) any observed anomalies or unsafe behaviors; and (iv) relevant environmental conditions like weather and road surface. Full prompt for Data Annotator can be found in appendix C.1.

The output of the Data Annotator is a structured, multi-faceted textual description for each clip. This descriptive summary serves as a standardized, information-rich input for the Traffic Scene Extractor and Driving Assessor agents in the subsequent stages of the workflow.

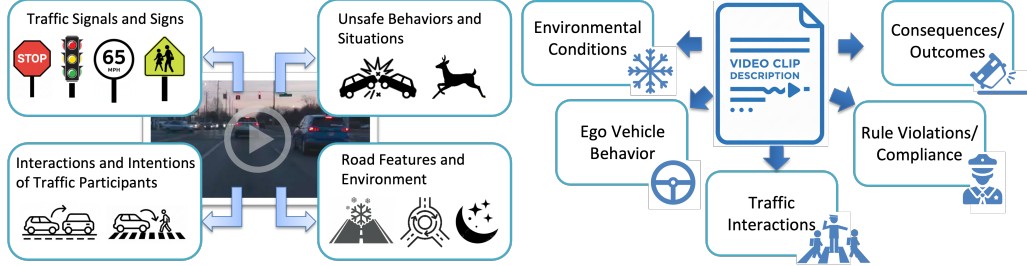

Figure 2: Analyze each video clip from four distinct and complementary perspectives.

Figure 3: A flexible set of scene types for LLM-based traffic scene extraction.

#### 3.3.2  TRAFFIC SCENE EXTRACTOR: NARRATIVE SCENE DECOMPOSITION

The Traffic Scene Extractor agent acts as an intelligent parser, taking the dense, multi-faceted narrative generated by the Data Annotator and distilling it into a set of discrete, safety-relevant scenes. While the Data Annotator provides a comprehensive overview, its raw output can be verbose and contain interwoven details from different moments or aspects of a clip. The Extractor's role is to streamline this information into focused, non-overlapping scene descriptions, which are critical for the precise knowledge retrieval and reasoning performed by downstream agents.

The core of the Traffic Scene Extractor is an LLM-based prompting strategy that guides the model to dissect the comprehensive annotation. The prompt in appendix C.2, as shown in figure 3, defines a flexible set of scene types, including Environmental Conditions, Ego Vehicle Behavior, Traffic Interactions, Rule Compliance/Violations, and Outcomes/Consequences. Crucially, the prompt instructs the LLM to incorporate causal factors, inferred intentions, participant interactions, and temporal se-

quences within each scene description. This ensures that the extracted scenes are not just isolated observations but semantically rich accounts of safety-critical moments. The prompt also includes detailed positive and negative examples of both normal driving and incident scenarios, demonstrating how to compose concise, safety-relevant descriptions and how to avoid redundant or irrelevant details. The primary goal is to extract up to 5 distinct safety-relevant scenes, each focusing on a specific aspect or moment from the annotation.This decomposition provides granular inputs for subsequent analysis. The output is a list of these refined textual scene descriptions, ready for use by the Traffic Rule Checker and Traffic Accident Retriever. This decomposition provides granular inputs for subsequent analysis. The output is a list of these refined textual scene descriptions, ready for use by the Traffic Rule Checker and Traffic Accident Retriever.

### 3.3.3 TRAFFIC ACCIDENT RETRIEVER: EXPERIENTIAL ANALYSIS

The Traffic Accident Retriever agent identifies latent risks by cross-referencing the current driving scene with a rich knowledge base of past incidents, capturing dangers that may not involve a formal rule violation. This agent provides the framework with an "experiential" understanding of what makes a situation hazardous, even if all parties are acting legally. The core of this component is an agentic GraphRAG pipeline that involves two major phases: offline knowledge base construction and online retrieval and analysis.

**Graph Construction**

The agent's knowledge is stored in the Historical Accident Graph Database ($K_A$), which is constructed from real-world incident data to serve as the system's long-term memory of dangerous scenarios. The process begins with a dataset of accident videos, such as the Nexar Dashcam Collision Prediction dataset, where 10-second clips containing collision or near-miss scenes are extracted. For each of these clips, the Data Annotator generates detailed textual annotations, which serve as the source documents for populating the graph. To ensure a consistent and structured representation of these incidents, we first define a graph schema in appendix D. This schema specifies explicit node labels like `Accident`, `TrafficCondition`, `TrafficAction`, and `TrafficParticipant`, along with the permissible relationships between them, such as `TrafficAction -[:CONTRIBUTE_TO]-> Accident` or `EgoVehicle -[:INTERACTS_WITH]-> TrafficParticipant`. The annotation documents are then processed through an automated pipeline to build the graph database (e.g., Neo4j). This involves chunking each annotation into smaller text segments and using a schema-constrained entity extraction tool, like LangChain's `LLMGraphTransformer`. This tool is configured with our predefined schema, guiding an LLM to extract entities and relationships that strictly conform to the allowed labels and types. These extracted entities are ingested as nodes with their descriptions, and the relationships form the edges of the graph. Additionally, vector embeddings are generated for each chunk and entity, enabling efficient semantic search later. This process is applied incrementally across all accident documents, culminating in a unified and interconnected knowledge graph of historical incidents.

**Agentic GraphRAG for Risk Retrieval**

During an evaluation, the Traffic Accident Retriever agent takes a structured scene description from the Traffic Scene Extractor as its query and performs a multi-stage Graph RAG retrieval process to find analogous historical incidents. This process is designed to maximize both relevance and recall. Initially, a hybrid search strategy is employed: the query is converted into an embedding for a vector similarity search across Chunk nodes in the graph, identifying semantically similar historical contexts. Concurrently, a full-text search is performed to capture chunks containing exact keyword matches. For the most relevant chunks identified by this hybrid approach, the agent then expands its search within the graph. It traverses 1-2 hops (adaptively based on similarity) from these chunks to retrieve all connected entities and their relationships, thereby gathering the full contextual details of the historical incident. This step moves beyond isolated text snippets to reconstruct complete accident scenarios from the graph.

The retrieved set of historical scenarios forms a candidate pool. A crucial subsequent step is agentic filtering, where an LLM assesses the contextual relevance of each candidate to the specific query scene, effectively filtering out any irrelevant or misleading results. Finally, the highly-relevant, filtered historical accident scenarios are passed to a dedicated LLM, prompted to act as a "traffic

accident analyst." This LLM, guided by the prompt (provided in appendix C.3), receives both the current traffic scene and the retrieved historical context. Its task is to synthesize this information to identify potential accidents that could plausibly arise from the current driving scene and to provide a clear, causal explanation for why the current scene is risky, based on the outcomes of similar past events. This sophisticated process allows the Traffic Accident Retriever to flag latent risks by effectively asking, "Have we seen a situation like this lead to an accident before?", thereby directly fulfilling the framework's principle of proactive, data-effective risk identification.

## 3.4 TRAFFIC RULE CHECKER: NORMATIVE ANALYSIS

The Traffic Rule Checker agent performs a normative analysis, auditing the driving behavior depicted in a scene against a verifiable knowledge base of codified traffic regulations. Before the online analysis, the agent's knowledge is prepared in the Traffic Rule Vector Database ($K_R$). This knowledge base is meticulously constructed by ingesting textual regulations from relevant jurisdictions, such as state-specific vehicle codes. Each rule is encoded into a high-dimensional vector using a sentence embedding model and stored in the Milvus database. This structure enables efficient semantic retrieval, allowing the agent to find relevant laws even if the query scene does not use exact legal terminology.

This agent is implemented as a robust, multi-step pipeline using a framework like LangGraph to ensure a structured and accurate evaluation. Its architecture combines an LLM for efficient reasoning with a Milvus vector database for rapid rule retrieval. The agent's core logic follows a sequential, three-stage agentic RAG process: Retrieve, Grade, and Verify. The agent's online workflow begins with the Retrieval step, where it takes a scene description from the Traffic Scene Extractor as a query. It performs a vector similarity search against the $K_R$ database to fetch the top-k most semantically similar traffic rules. However, semantic similarity alone can sometimes retrieve rules that are related but not directly applicable. To address this and prevent false positives, the process moves to the Relevance Grading step. Here, an LLM acts as a grader, analyzing each retrieved rule in the context of the specific driving scene. Guided by a the prompt in appendix C.4, it filters out any rules deemed irrelevant, ensuring that only the most pertinent statutes are passed to the final stage. This critical filtering step significantly reduces noise and focuses the final analysis.

In the final Violation Verification step, the original scene description and the filtered, highly-relevant rules are passed to another LLM. This LLM is prompted with a specific persona, such as a "police officer," to meticulously analyze the scene and determine if the vehicle's actions violated any of the provided rules. The agent's final output is a structured object, conforming to a Pydantic model, which clearly states whether a violation was "found" or "not_found" and provides a concise reason for the judgment. This multi-step, agentic process of retrieving, grading, and verifying ensures that the final assessment of legal compliance is not only accurate but also robust and contextually grounded.

## 3.5 DRIVING ASSESSOR: HOLISTIC SYNTHESIS AND REPORTING

The Driving Assessor is the final agent in the DriveEval pipeline, responsible for synthesizing the analytical outputs from all upstream agents into a single, cohesive, and actionable safety report ($R$). This agent functions as an expert driving instructor and safety analyst, leveraging a powerful LLM to perform a final, holistic evaluation of the driving performance observed in a given scene.

The agent's reasoning process is initiated by providing the LLM with a comprehensive set of inputs for each analyzed clip. These inputs include the rich textual scene description from the Data Annotator, the list of identified violations from the Traffic Rule Checker, and the analogous historical risks surfaced by the Traffic Accident Retriever. This aggregated information is structured into a detailed prompt in appendix C.5, which guides the LLM to perform its analysis in a structured and evidence-based manner.

A core component of this agent's methodology is a standardized safety scoring rubric, which anchors the LLM's evaluation to a predefined set of criteria. This rubric prevents subjective or arbitrary assessments and ensures consistency across all evaluations. The scoring is based on a 1-10 scale, which is mapped to four distinct risk levels: Critical (scores 1-4), High (5-7), Medium (8), and Low (9-10). The specific criteria for each score are highly contextual, depending on whether an

accident or near-miss occurred, who was at fault (the ego-vehicle or another participant), and the quality of the ego-vehicle's reaction or attempt to mitigate the situation. For example, scores in the "Critical" range are reserved for actual collisions, with a lower score indicating greater fault on the part of the ego-vehicle. This detailed, evidence-based rubric forces the LLM to move beyond simple summarization and perform nuanced causal reasoning.

After applying the rubric to determine the most appropriate safety score, the agent is prompted to generate the final, multi-faceted report ($R$) as defined in our Problem Formulation. This structured output includes: the final safety score and its corresponding risk level; a holistic narrative evaluation that justifies the score by summarizing the key events and behaviors; distinct lists of identified strengths and weaknesses to provide balanced feedback; and a set of specific, actionable improvement advice designed to guide developers in rectifying the identified flaws. This final synthesis step transforms a collection of isolated analytical findings into a valuable and interpretable evaluation, directly fulfilling the framework's core principle of producing explainable and actionable results.

## 4 EXPERIMENTS AND CASE STUDIES

### 4.1 EXPERIMENTAL SETUP

To rigorously assess the performance of the DriveEval framework and its constituent agents, we conducted a series of experiments focusing on the effectiveness of various Large Language Models (LLMs) and Vision-Language Models (VLMs) within their respective roles. This section details the datasets employed, the specific models chosen for evaluation, and the infrastructure supporting our experimental setup.

**Datasets** Our evaluation strategy is built upon two categories of datasets: 1) the primary evaluation dataset **DriveEval**, which provides comprehensive system assessment through 200 diverse dashcam clips (about 35 minutes) paired with human-annotated safety reports created via a four-step process of query-driven video sourcing, manual event marking, precise clip extraction, and structured report generation; 2) and the knowledge base datasets, which ground the agent through two components: the **Traffic Rule Vector Database** ($K_R$), derived from the Pennsylvania Driver's Manual by converting it into Markdown, segmenting, and embedding for semantic retrieval; and the **Historical Accident Graph Database** ($K_A$), constructed from 750 accident or near-miss videos in the Nexar dataset, where annotated 10-second clips were processed into textual descriptions, organized under a manually defined schema, and transformed into a Neo4j knowledge graph capturing entities and relationships. For more details, please see A.

**Models for Evaluation** Our evaluation methodology focuses on assessing the performance of various state-of-the-art Large Language Models (LLMs) and Vision-Language Models (VLMs) within each agent's specific role. For conciseness, all model names are presented in a simplified format, omitting provider and platform prefixes (e.g., 'gpt-4o' for 'openai:gpt-4o'). **Data Annotator (VLM):** This agent, responsible for initial scene interpretation, was evaluated using the following VLMs, which process frame image sequences from video clips: 'gpt-4o', 'gpt-4.1', 'gpt-5', 'o3', 'claude-opus-4-1', 'claude-sonnet-4-2', 'gemini-2.5-pro', 'gemma-3-27b', 'grok-4', 'qwen2-5-vl-72b-instruct', 'qwen-vl-max', and 'glm-4.5v'. **Other Agents (LLM):** For the Traffic Scene Extractor, Traffic Rule Checker, Traffic Accident Retriever, and Driving Assessor agents, which primarily handle textual inputs, we evaluated a comprehensive set of LLMs including: 'gpt-4o', 'gpt-4.1', 'gpt-5', 'o3', 'claude-opus-4-1', 'claude-sonnet-4-2', 'gemini-2.5-pro', 'gemma-3-27b', 'grok-4', 'qwen2-5-vl-72b-instruct', 'qwen-vl-max', 'glm-4.5v', 'llama-3.3-70b-versatile', 'deepseek-r1-distill-llama-70b', 'qwen3-max', 'glm-4.5', 'sonar-pro', 'qwen3-32b', 'gpt-oss-120b', 'kimi-k2-instruct-0905', 'llama-4-scout-17b-16e-instruct', and 'llama-4-maverick-17b-128e-instruct'.

**Implementation Details** The DriveEval framework is orchestrated using LangGraph, which manages the multi-agent pipeline and their interactions. For the underlying data storage and retrieval, Milvus serves as the vector database for $K_R$, while Neo4j is utilized as the graph database for $K_A$.

### 4.2 DISCUSSION ON BEST MODEL FOR AGENTS

We explore the best model for each agent and the results are provided in Appendix E. For data annotator agent, gpt-4o can achieve best performance on traditional similarity metrics while consuming

Figure 4: An aggressive and illegal maneuver by another vehicle

least time. While for traffic scene extractor agent, llama-3.3-70b obtains highest scores on textual metrics but is almost 10x slower than fastest model. A similar phenomenon can also be observed in other agents. Thus, the best model for each agent does not remain consistent, which further validates the necessity of multi-agent design, which assigns the optimal LLM to the corresponding agent.

## 4.3 CASE STUDY

We make two case studies to show how our DriveEval framework works in real-world scenes:

**Case 1:** To demonstrate DriveEval's capabilities, we analyze a complex near-miss event from our dataset involving an aggressive merge from the shoulder, as shown in figure 4. The ego-vehicle is traveling in the rightmost lane of a highway when a black pickup truck accelerates rapidly along the right shoulder and merges abruptly into the ego-vehicle's lane directly in front of it, without signaling. The Data Annotator successfully captures these key details, producing a rich description that notes the "abrupt merge," the "lack of signaling," and the resulting "unsafe" and "close following distance." The Traffic Scene Extractor then decomposes this narrative into distinct, safety-relevant scenes for analysis, including "a black pickup truck merging into the ego-vehicle's lane from the right shoulder without signaling" and "ego-vehicle maintains its lane and speed, following the pickup truck at a close distance after the merge, resulting in a reduced following distance." Traffic Rule Checker correctly identifies the pickup truck's multiple violations, such as "failure to yield to traffic already on the major roadway," "not following the steps to merge with traffic from an acceleration lane," and "merging from the shoulder without signaling. Traffic Accident Retriever finds that the current scenario strongly resembles past incidents, leading to a side-swipe and a rear-end collision. The Driving Assessor synthesizes these findings into a comprehensive safety report. It provides improvement advice, that the ego vehicle should "immediately adjust your speed and increase your following distance" and "stay alert for vehicles on the shoulder." These system results are highly aligned with the human's judgment.

**Case 2:** This case study highlights the critical importance of geographical context for accurate traffic rule checking. In a scenario where the ego-vehicle operates in a left-hand traffic region and veers slightly left to avoid a head-on collision with an oncoming vehicle occupying its lane, the Data Annotator accurately describes the evasive action. However, if the Traffic Rule Checker is powered by a knowledge base ($K_R$) designed for right-hand traffic, it incorrectly flags a violation, stating that the leftward maneuver goes against the rule to "escape to the right if possible." This false positive directly impacts the Driving Assessor, which would then assign a suboptimal safety score. This scenario starkly illustrates DriveEval's need for adaptive, region-specific knowledge bases that can dynamically switch or modify traffic rules based on the operational context, ensuring accurate normative analysis across diverse global driving conventions.

## 5 CONCLUSION AND FUTURE WORK

In this paper, we proposed DriveEval, a context-aware multi-agent framework for evaluating the safety of autonomous driving systems. By coordinating agents for scene parsing, rule checking, and accident retrieval, the framework leverages language and vision-language models to provide transparent and fine-grained assessments that complement traditional quantitative metrics. Our experiments demonstrate that DriveEval not only aligns well with human judgment but also highlights subtle risks and rare corner cases often overlooked by existing evaluation pipelines. For future work, we aim to scale evaluations to larger and more diverse datasets to ensure robustness across regions, weather, and traffic conditions. Furthermore, closing the loop between evaluation and training through automatic scenario generation and curriculum design can accelerate system improvement.

ETHICS STATEMENT

This work focuses on advancing the safety evaluation of autonomous driving (AD) through context-aware multi-agent methods. Our framework is intended solely for research and safety assurance, not for deployment without rigorous validation. We acknowledge that reliance on large language and vision-language models may introduce biases from training data, which could affect fairness or reliability across regions. To mitigate such risks, we emphasize transparency through interpretable, provenance-backed assessments and encourage further audits before real-world adoption. We also recognize broader societal impacts: while improving AD safety can reduce accidents and save lives, misuse or premature deployment of evaluation tools may create false confidence. Responsible use therefore requires careful collaboration with regulators, domain experts, and affected communities.

REPRODUCIBILITY STATEMENT

We provide detailed descriptions of our framework design, agent roles, datasets, and evaluation protocols in the main text and appendix. All experiments rely on publicly available datasets and models, with hyperparameters, prompts, and implementation details specified to ensure replicability. Code has been provided in anonymous Github repository `https://anonymous.4open.science/r/DriveGuard-811C/` to facilitate verification and extension by the community.

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

## A    Supplementary Introduction on Datasets

Our evaluation strategy is underpinned by two categories of datasets: a primary ground-truth evaluation dataset for comprehensive system assessment, and specialized knowledge base datasets for agent grounding.

**Primary Evaluation Dataset (DriveEval)**    For a rigorous, in-depth evaluation of our framework, we developed the DriveEval Dataset, a novel collection of 200 diverse dashcam video clips (totaling approximately 35 minutes) paired with comprehensive, human-annotated safety reports. The dataset was created through a robust 4-step workflow: (1) building various queries and sourcing raw videos from YouTube using over 175 targeted, safety-critical search queries; (2) using an interactive web-based tool for human annotators to manually review and mark relevant events; (3) extracting these marked events into precisely timed 10-second clips; and (4) generating gold-standard, structured ground-truth reports for each clip using a dedicated annotation tool that mirrors our framework's final output.

**Knowledge Base Datasets    Normative Knowledge** ($K_R$): The Traffic Rule Vector Database was constructed by ingesting the official Pennsylvania Driver's Manual (a PDF document). This manual was first converted to Markdown, subsequently segmented into manageable chunks, and then embedded into a Milvus vector database to facilitate efficient semantic retrieval of specific traffic regulations. **Experiential Knowledge** ($K_A$)**:** The Historical Accident Graph Database was built upon the Nexar Dashcam Crash Prediction dataset. We utilized all approximately 750 videos classified as "positive" (i.e., containing a collision or an imminent near-miss). From these, 10-second clips capturing the full accident or near-miss event were extracted. Our Data Annotator processed these clips to generate detailed traffic annotations. Based on these unstructured textual descriptions, we manually defined a comprehensive graph schema. The Neo4j LLM Knowledge Graph Builder was then employed to construct the knowledge graph in Neo4j, extracting entities and relationships from these annotations in accordance with our predefined schema.

## B    Statement on the Use of Large Language Models (LLMs)

In the spirit of transparency and in accordance with conference guidelines, we disclose the use of Large Language Models (LLMs) as assistive tools in the preparation of this paper and its associated software.

Throughout the preparation of this manuscript, we utilized LLMs, specifically Gemini 2.5 Pro and GPT-5, as writing assistants. Their role was primarily confined to improving the clarity, flow, and grammatical correctness of the text. The models were used for tasks such as rephrasing sentences for better readability, correcting grammar, and polishing the overall prose. However, the core scientific ideation, the formulation of the methodology, the analysis of results, and the final conclusions were conceived and articulated entirely by the human authors.

In addition to writing, the Claude Code was employed as a coding assistant during the code development phase of this project. Its role was significant in accelerating the implementation of our custom dataset collection and annotation tools. Furthermore, Claude was used to assist in debugging and refining the project's codebase. All AI-generated code was thoroughly reviewed, tested, and validated by the authors to ensure its correctness and alignment with the project's objectives.

We affirm that LLMs were used as productivity tools and are not considered authors of this work. The human authors take full responsibility for the scientific integrity, accuracy, and originality of all content, including any text or code initially suggested by an LLM.

## C    Full Prompts

### C.1    Prompt for Data Annotator

The following is the complete prompt used for the Data Annotator agent, designed to guide the Vision-Language Model (VLM) in generating rich, multi-faceted textual descriptions from dashcam video clips.

You are an expert in analyzing dashcam footage for driving behavior
    research.
Analyze this dashcam video clip for indexing in a driving behavior
    analysis system.  Focus on concisely describing key elements relevant
     to driving safety, interactions, and intentions, with particular
    attention to but not limited to the following:

**Traffic Signals and Traffic signs:**
* Describe the following aspects throughout the clip, noting any
    transitions. Describe the traffic signs. Describe the the ego-vehicle
    's actions in response to each change.
  - Road signs: Speed limits, stop signs, yield signs, warning signs, etc
    .
  - Traffic signals: Red, yellow, green lights, pedestrian signals.
  - Other Lane markings: Yellow and white solid lines, dashed lines, bus
      lanes, bike lanes, left-turn only, right-turn only, marked
      crosswalk, etc.

**Interactions and Intentions of Traffic Participants:**
* Describe the location/position of the ego-vehicle (first lane to the
    left, closest opposite lane, etc.), given the opposite lines are
    usually divided by solid yellow lines and others are divided by white
     lanes.
* Identify all visible traffic participants, their types(Car, truck, bus,
     motorcycle, bicycle, pedestrian, etc.), and attributes (Size, color,
     make/model, lane position, speed, direction, signaling). Note any
    vulnerable road users (VRUs), such as pedestrians, cyclists, and
    motorcyclists.
* Describe the sequence of events in the clip, paying close attention to
    the timing of actions and reactions between the ego-vehicle and other
     traffic participants.
* Describe the movements of other vehicles, particularly those that turn
    left or right, merge into the ego-vehicle's lane, or change lanes
    abruptly. Note their trajectories relative to the ego-vehicle.
* Describe the interactions and intentions of traffic participants,
    including the ego-vehicle. Analyze the following aspects:
  - Following distance: Is the ego-vehicle maintaining a safe following
      distance?
  - Lane changes: Are lane changes executed safely and with proper
      signaling?
  - Merging/yielding: How does the ego-vehicle handle merging and
      yielding situations?
  - Right-of-way: Does the ego-vehicle respect right-of-way rules?
  - Turning: Analyze turning maneuvers for smoothness, signal use, and
      adherence to lane markings.
* Analyze and explain the reasons or intentions of above behaviors if any
    .

**Unsafe Behaviors and Situations:**
* Identify any situations where the ego-vehicle or other vehicles come
    close to each other, such as near-collisions, sudden braking, or
    sharp turns. Describe the factors that contributed to these
    situations.
* Identify any traffic violations by the ego-vehicle or other
    participants, such as running red lights, speeding, illegal lane
    changes, or failing to yield.
* Other unsafe behaviors:
  - Speeding: Is the ego-vehicle exceeding the speed limit or driving too
      fast for conditions?
  - Aggressive driving: Tailgating, weaving through traffic, sudden
      braking, wrong way, etc.
  - Distracted driving: Any signs of the driver being distracted (e.g.,
      phone use, eating, looking away from the road).
  - Drowsy driving: Any indications of driver fatigue.

```
* Unusual Circumstances:
  - Construction zones
  - Accidents
  - Emergency vehicles
  - Road debris
  - Pedestrians or animals unexpectedly entering the roadway
* Note please skip the description of the absence of findings.

**Road Features and Environment:**
* Describe relevant road types(intersection, roundabouts, local/
    residential roads, highways, freeways, expressways, one-way roads,
    etc.), features(single lane, multiple lanes, road conditions, etc.),
    and conditions(Dry, wet, icy, slippery, etc.).
* Note the environmental context [weather(Sunny, cloudy, rainy, snowy,
    foggy, etc.), time of day(Daytime, nighttime, dawn, dusk, etc.),
    visibility(Clear, limited, etc.)].

Provide an informative description, avoid statements about the absence of
    findings.
```

## C.2    PROMPT FOR TRAFFIC SCENE EXTRACTOR

The following is the complete prompt used for the Traffic Scene Extractor agent, designed to guide the Large Language Model in decomposing dashcam annotations into distinct, safety-relevant scenes.

```
[
    ("system", "You are a traffic scene decomposition expert. Your task
        is to extract driving safety scenes from dashcam annotations,
        focusing only on aspects relevant to traffic rule checking and
        accident risk assessment."),
    ("user", """
    Extract UP TO 5 distinct driving safety scenes from the dashcam
        annotation. Focus only on scenes relevant to driving safety -
        skip irrelevant aspects.

    **FLEXIBLE SCENE TYPES** (use as needed, can repeat important types):

    1. ENVIRONMENTAL CONDITIONS
       - Road layout, weather, visibility, traffic density affecting
           safety
       - Example: "Four-lane bridge with solid yellow no-passing lines
           during clear daylight conditions"

    2. EGO VEHICLE BEHAVIOR
       - Ego vehicle's driving actions, decisions, speed, positioning
       - Example: "Ego vehicle maintains safe following distance while
           traveling in left lane of bridge"

    3. TRAFFIC INTERACTIONS
       - Interactions between vehicles, normal or risky
       - Example: "Blue car encounters stopped vehicle and begins evasive
            lane change maneuver"

    4. RULE COMPLIANCE/VIOLATIONS
       - Following traffic rules properly OR breaking them
       - Example: "Blue car crosses solid yellow no-passing zone lines
           into oncoming traffic lane"

    5. **OUTCOMES/CONSEQUENCES**
       - Results of actions: safe completion or negative consequences
       - Example: "Head-on collision causes ego vehicle to lose control
           and leave roadway"
```

```
**RICH CONTEXT GUIDELINES** (include when available):
- **CAUSES**: What led to this situation (e.g., stopped car blocking
    lane)
- **INTENTIONS**: Why actions were taken (e.g., avoiding obstacle,
    reaching destination)
- **INTERACTIONS**: How participants affect each other (e.g., forcing
    sudden braking)
- **SEQUENCES**: Order of events (e.g., lane change -> crossing lines
    -> collision)
- **CONDITIONS**: Environmental factors affecting behavior (e.g., wet
    roads, limited visibility)

**FLEXIBLE COMPOSITION EXAMPLES:**

**Normal Driving Scenario:**
- "Four-lane residential street with 25mph speed limit during clear
    afternoon conditions, moderate traffic density with parked cars
    lining both sides creating narrow travel lanes"
- "Ego vehicle maintains appropriate 22mph speed in right lane,
    positioned center of travel lane with 3-second following distance
     behind silver sedan, driver demonstrating cautious behavior due
    to parked car obstacles"
- "Pedestrian approaches marked crosswalk from right sidewalk; ego
    vehicle recognizes pedestrian's intention to cross, begins
    gradual deceleration 50 feet before crosswalk, comes to complete
    stop to yield right-of-way"
- "Ego vehicle activates right turn signal 100 feet before
    residential driveway entrance, checks mirrors for cyclists,
    reduces speed to 8mph for safe turning radius while ensuring no
    oncoming traffic conflicts"
- "Successful completion of residential navigation with consistent
    rule compliance, appropriate speed management for conditions, and
     proactive safety measures protecting vulnerable road users"

**Incident Scenario:**
- "Two-lane bridge with solid yellow no-passing lines separating
    opposing traffic, clear weather but limited escape routes due to
    concrete barriers, 45mph speed limit with moderate traffic flow"
- "Blue sedan in opposite direction encounters stopped disabled
    vehicle blocking its travel lane, driver attempts emergency lane
    change but misjudges available space and oncoming traffic speed,
    panic response leads to overcorrection"
- "Blue sedan crosses solid yellow no-passing zone markings into ego
    vehicle's lane while traveling approximately 40mph, violating
    traffic law prohibiting passing in no-passing zone, creating
    immediate head-on collision risk"
- "Head-on collision occurs as blue sedan strikes ego vehicle's front
    -left quarter panel, impact forces cause ego vehicle to lose
    directional control despite driver's attempted evasive steering"
- "Ego vehicle crosses into oncoming lanes and impacts concrete
    barrier before coming to rest, collision sequence demonstrates
    how improper passing decisions escalate from a traffic violation
    to severe multi-vehicle incident with potential for serious
    injuries"

**OUTPUT FORMAT:** Return clean scene descriptions without scene
    numbers or type labels. Focus purely on the safety-relevant
    content.

**ANTI-OVERLAP RULES:**
- Each scene focuses on ONE specific aspect or moment
- Avoid describing the same incident from multiple perspectives
- Important safety aspects can have multiple scenes if distinct
- Skip scene types not relevant to driving safety
```

```
    Complex annotation:
    {annotation}

    Extract up to 5 distinct safety-relevant scenes. Include rich context
        (causes, intentions, interactions, sequences) when available.
        Focus each scene on one specific aspect.
    """)
]
```

### C.3 PROMPT FOR ACCIDENT RETRIEVER

The following is the complete prompt used for identifying potential accidents.

```
[
    ("system", "You are a traffic accident analyst. You are given
        historical traffic accidents retrieved from neo4j graph database,
         and you need to summarize the possible accidents of the traffic
        scene."),
    ("user", """
    The historical traffic accidents are inside the <
        historical_traffic_accidents> tag. The traffic scene is inside
        the <traffic_scene> tag.

    ### Response Guidelines:
    1. **Direct Answers**: Provide clear and thorough answers to the user
        's queries without headers unless requested. Avoid speculative
        responses.
    2. **Utilize History and Context**: Leverage relevant information
        from the current driving scene, and the context provided below.
    3. **No Greetings in Follow-ups**: Start with a greeting in initial
        interactions. Avoid greetings in subsequent responses unless
        there's a significant break or the chat restarts.
    5. **Avoid Hallucination**: Only provide information based on the
        context provided. Do not invent information.
    6. **Response Length**: Keep responses concise and relevant. Aim for
        clarity and completeness within 4-5 sentences unless more detail
        is requested.
    7. **Tone and Style**: Maintain a professional and informative tone.
        Be friendly and approachable.
    8. **Error Handling**: If a query is ambiguous or unclear, ask for
        clarification rather than providing a potentially incorrect
        answer.
    10. **Context Availability**: If the context is empty, do not provide
         answers based solely on internal knowledge. Instead, respond "No
         possible accident is found."

    ### Answer Format:
    - Possible accidents
    - Explanation reasons why the current driving scene can cause the
        accidents

    <traffic_scene>
    {traffic_scene}
    </traffic_scene>

    <historical_traffic_accidents>
    {historical_traffic_accidents}
    </historical_traffic_accidents>
    """)
]
```

### C.4 PROMPT FOR RULE CHECKER

```
[
    ("system", "You are a retrieval grader for traffic rule retrieval.
        Given a query, you are grading the relevance of the retrieved
        traffic rule.."),
    ("user","""
    The query is inside the <query> tag, and the retrieved traffic rule
        is inside the <retrieved_traffic_rule> tag.

    <query>
    {query}
    </query>

    <retrieved_traffic_rule>
    {retrieved_traffic_rule}
    </retrieved_traffic_rule>

    The goal is to filter out the retrieved traffic rule that is not
        relevant to the query, so grade it as not relevlant only if the
        retrieved traffic rule cannot provide any information to
        determine if the query violates the traffic rule or not.
    If the retrieved traffic rule contains some information that can be
        used to determine if the query violates the traffic rule or not,
        even if it is not the exact answer, then grade it as relevant.
    Output 'no' if the retrieved traffic rule is completely not relevant
        to the query, otherwise output 'yes'.
    """
    )
]
```

## C.5 PROMPT FOR DRIVING ASSESSOR

```
[
    ("system", """You are an expert driving instructor and safety analyst
        . Your role is to provide comprehensive, constructive feedback on
         driving behavior based on traffic scene analysis, accident risks
        , and rule violations.

Your assessment should be:
- Objective and evidence-based using the standardized safety scoring
    criteria
- Constructive and educational
- Focused on safety improvement
- Specific and actionable"""),
    ("user", """
    Analyze the following driving scenario and provide a comprehensive
        safety assessment:

    **Complex Traffic Annotation:**
    {annotation}

    **Accident Analysis Results:**
    {accident_results}

    **Traffic Rule Violation Results:**
    {rule_results}

    ## SAFETY SCORING CRITERIA (1-10)

    Use these specific criteria to assign the safety score. Choose the
        score that best matches the observed scenario:

    **CRITICAL RISK LEVELS (1-4):**
```

```
    - **Score 1**: Accident involving the ego vehicle caused by ego
        vehicle's traffic rule violation
    - **Score 2**: Accident involving the ego vehicle caused by ego
        vehicle's fault/risky behavior (non-violation)
    - **Score 3**: Accident involving the ego vehicle caused by others,
        but ego vehicle did not react properly or failed to mitigate
    - **Score 4**: Accident involving the ego vehicle caused by others,
        ego vehicle tried best to mitigate damages/loss

    **HIGH RISK LEVELS (5-7):**
    - **Score 5**: Near miss involving the ego vehicle caused by ego
        vehicle's traffic rule violation
    - **Score 6**: Near miss involving the ego vehicle caused by ego
        vehicle's risky behavior (non-violation)
    - **Score 7**: Near miss involving the ego vehicle caused by other
        traffic participants, ego vehicle involved but not at fault

    **MEDIUM RISK LEVEL (8):**
    - **Score 8**: No accident or near miss, but ego vehicle violates
        traffic rules or exhibits risky behaviors

    **LOW RISK LEVELS (9-10):**
    - **Score 9**: No accident or near miss, other traffic participants
        violate rules or exhibit risky behaviors, ego vehicle takes
        defensive actions
    - **Score 10**: Safe driving with good behavior, defensive driving,
        correct response to emergencies

    ## RISK LEVEL MAPPING
    - **Critical**: Scores 1-4 (Actual accidents involving the ego
        vehicle occurred)
    - **High**: Scores 5-7 (Near misses involving the ego vehicle
        occurred)
    - **Medium**: Score 8 (Violations/risky behavior without immediate
        danger)
    - **Low**: Scores 9-10 (Safe or defensive driving)

    ## ASSESSMENT REQUIREMENTS

    Based on the above criteria, provide:

    1. **Safety Score (1-10)**: Rate using the exact criteria above -
        justify your score selection
    2. **Overall Evaluation**: Summarize the driving performance with
        reference to the scoring criteria
    3. **Strengths**: Identify positive behaviors, defensive actions, and
        proper emergency responses
    4. **Weaknesses**: Point out risky behaviors, violations, poor
        reactions, or missed opportunities for safety
    5. **Improvement Advice**: Provide specific, actionable
        recommendations to move toward higher safety scores
    6. **Risk Level**: Assign based on score mapping (critical/high/
        medium/low)

    ## SCORING GUIDELINES:
    - **Accidents take precedence**: Any actual collision/accident
        involving the ego vehicle = scores 1-4 regardless of other
        factors
    - **Near misses are serious**: Close calls involving the ego vehicle
        without contact = scores 5-7
    - **Distinguish fault**: Consider who caused the incident (ego
        vehicle vs others)
    - **Evaluate response**: How well did ego vehicle react to others'
        mistakes?
```

```
      - **Context matters**: Consider weather, visibility, traffic density,
          road conditions
      - **Defensive driving**: Reward proactive safety measures and
          anticipation
      - **Multiple incidents**: Use the lowest applicable score if multiple
          safety issues occur

      ## OUTPUT FOCUS:
      - Be specific about which scoring criteria applies
      - Reference specific moments in the annotation
      - Explain the gap between current score and higher safety levels
      - Provide constructive rather than punitive feedback
      - Help driver understand why certain behaviors affect safety scores
      """)
]
```

# D ACCIDENT GRAPH SCHEMA

The following is the customized schema we defined for accident annotation documents.

```
TrafficCondition -[:CONTRIBUTE_TO]-> Accident
TrafficAction -[:CONTRIBUTE_TO]-> Accident
TrafficRuleViolation -[:CONTRIBUTE_TO]-> Accident

Accident -[:INCLUDES]-> RollOver
Accident -[:INCLUDES]-> PotentialCollision
Accident -[:INCLUDES]-> NearMiss
Accident -[:INCLUDES]-> UnsafeSituation
Accident -[:INCLUDES]-> Collision
Accident -[:INCLUDES]-> Injury

LaneMarking -[:INDICATES]-> LaneType
LaneType -[:IMPACTS]-> TrafficCondition
TrafficSignal -[:IMPACTS]-> TrafficCondition
RoadSign -[:IMPACTS]-> TrafficCondition
RoadFeature -[:IMPACTS]-> TrafficCondition
Environment -[:IMPACTS]-> TrafficCondition

TrafficAction -[:CONTRIBUTE_TO]-> TrafficRuleViolation
TrafficCondition -[:CONTRIBUTE_TO]-> TrafficRuleViolation

TrafficParticipant -[:Respond_To]-> TrafficSignal
EgoVehicle -[:Respond_To]-> RoadSign

EgoVehicle -[:INTERACTS_WITH]-> TrafficParticipant
TrafficParticipant -[:INTENDS]-> TrafficAction
EgoVehicle -[:INTENDS]-> TrafficAction
EgoVehicle -[:EXECUTES]-> TrafficAction
TrafficParticipant -[:EXECUTES]-> TrafficAction
```

# E EVALUATION TABLES

## E.1 DATA ANNOTATOR

**Metrics Overview.** The annotation component evaluates multimodal models on their ability to generate comprehensive driving scenario descriptions. **Traditional similarity metrics** include BLEU and ROUGE-L, which measure lexical and sequential overlap, and Semantic Similarity, which captures meaning beyond word overlap using embeddings. **LLM-as-Judge metrics** (scored 1–10) include Accuracy, measuring factual correctness of events; Completeness, reflecting coverage of critical driving events; and Clarity, assessing readability and utility of annotations for downstream analysis.

Table 1: Annotation Component performance across multimodal models. Best values in each column are highlighted in bold.

| Model | BLEU | ROUGE-L | Semantic | Accuracy | Completeness | Clarity | Avg Time |
|---|---|---|---|---|---|---|---|
| gpt-4o | **0.43** | **0.63** | **0.91** | 6 | 5 | 8 | **12.9s** |
| gpt-4.1 | 0.10 | 0.29 | 0.81 | 6 | 7 | **9** | 29.8s |
| gpt-5 | 0.03 | 0.21 | 0.78 | **8** | **8** | **9** | 1m 50s |
| o3 | 0.01 | 0.16 | 0.67 | **8** | **8** | **9** | 41.3s |
| claude-opus-4-1 | 0.07 | 0.24 | 0.76 | 5 | 4 | 8 | 17.9s |
| claude-sonnet-4 | 0.05 | 0.24 | 0.77 | 5 | 4 | 8 | 22.6s |
| gemini-2.5-pro | 0.10 | 0.27 | 0.74 | 6 | 6 | 8 | 19.4s |
| gemma-3-27b | 0.09 | 0.25 | 0.73 | 4 | 3 | 7 | 21.1s |
| grok-4 | 0.06 | 0.29 | 0.82 | 5 | 5 | 7 | 39.8s |
| qwen2.5-vl-72b | 0.11 | 0.30 | 0.81 | 4 | 3 | 7 | 19.7s |
| qwen-vl-max | 0.08 | 0.26 | 0.77 | 5 | 5 | 8 | 22.4s |
| glm-4.5v | 0.06 | 0.29 | 0.85 | 5 | 4 | 8 | 28.4s |

## E.2 TRAFFIC SCENE EXTRACTOR

**Metrics Overview.** We evaluate scene extraction performance using three categories of metrics. **Traditional similarity metrics** include BLEU and ROUGE-L, which measure lexical and sequential overlap with human-written scenes, as well as Semantic Similarity, which captures meaning beyond word overlap using embeddings. **Enhanced averages** include Safety Avg, which weights detection toward safety-critical events (e.g., accidents, near-misses), Temporal Avg, which evaluates chronological consistency of scene ordering, and Coherence Avg, which measures narrative flow across transitions. **LLM-as-Judge scores** (on a 1–10 scale) include Extraction Quality, assessing boundary detection and completeness, Temporal Coherence, measuring logical ordering, and Safety Relevance, evaluating the focus on safety-critical content.

Table 2: Scene Component performance across text models. Best values in each column are highlighted in bold.

| Model | BLEU | ROUGE-L | Semantic | Coverage | Safety Avg | Temporal Avg | Coherence Avg | Extract | Temporal | Safety | Avg Time |
|---|---|---|---|---|---|---|---|---|---|---|---|
| gpt-4o | 0.20 | 0.40 | 0.86 | 0.96 | 0.50 | 0.64 | 0.69 | 9 | 10 | 10 | 11.0s |
| gpt-4.1 | 0.14 | 0.37 | 0.87 | 0.96 | 0.64 | 0.66 | 0.68 | 9 | 10 | 10 | 4.9s |
| gpt-5 | 0.08 | 0.30 | 0.84 | 0.95 | 0.47 | 0.71 | 0.66 | 9 | 10 | 10 | 22.6s |
| o3 | 0.05 | 0.26 | 0.83 | 0.95 | 0.48 | 0.54 | 0.67 | 9 | 10 | 10 | 11.5s |
| claude-opus-4-1 | 0.05 | 0.27 | 0.84 | 0.94 | 0.54 | 0.72 | 0.71 | 9 | 10 | 10 | 9.5s |
| claude-sonnet-4 | 0.07 | 0.29 | 0.86 | 0.94 | 0.61 | 0.77 | 0.74 | 9 | 10 | 10 | 11.7s |
| gemini-2.5-pro | 0.08 | 0.29 | 0.82 | 0.87 | 0.51 | 0.78 | 0.72 | 9 | 10 | 10 | 15.8s |
| gemma-3-27b | 0.10 | 0.35 | 0.85 | 0.95 | 0.55 | 0.79 | 0.66 | 9 | 10 | 10 | 2.9s |
| grok-4 | 0.09 | 0.31 | 0.86 | 0.96 | 0.59 | 0.78 | 0.71 | 9 | 10 | 10 | 19.0s |
| qwen2.5-vl-72b | 0.12 | 0.34 | 0.86 | 0.95 | 0.46 | 0.79 | 0.74 | 9 | 10 | 10 | 9.6s |
| qwen-vl-max | 0.09 | 0.32 | 0.85 | 0.96 | 0.52 | 0.74 | 0.72 | 9 | 10 | 10 | 4.8s |
| glm-4.5v | 0.09 | 0.32 | 0.84 | 0.94 | 0.54 | 0.80 | 0.68 | 9 | 10 | 10 | 30.2s |
| llama-3.3-70b | **0.41** | **0.57** | **0.90** | **0.96** | 0.64 | 0.79 | 0.72 | 9 | 10 | 10 | 5.2s |
| deepseek-r1-70b | 0.10 | 0.36 | 0.83 | 0.94 | 0.49 | 0.74 | 0.60 | 9 | 10 | 10 | 4.1s |
| qwen3-max | 0.05 | 0.25 | 0.84 | 0.96 | 0.64 | 0.75 | 0.69 | 9 | 10 | 10 | 5.5s |
| glm-4.5 | 0.08 | 0.30 | 0.86 | 0.95 | 0.51 | 0.74 | 0.72 | 9 | 10 | 10 | 12.5s |
| sonar-pro | 0.13 | 0.35 | 0.86 | 0.92 | 0.58 | 0.75 | 0.70 | 9 | 10 | 10 | 6.7s |
| qwen3-32b | 0.09 | 0.32 | 0.86 | 0.94 | 0.60 | 0.74 | 0.68 | 9 | 10 | 10 | 2.8s |
| gpt-oss-120b | 0.08 | 0.32 | 0.86 | 0.96 | 0.45 | 0.78 | 0.68 | 9 | 10 | 10 | 3.1s |
| kimi-k2 | 0.07 | 0.30 | 0.85 | 0.96 | 0.60 | 0.73 | 0.68 | 9 | 10 | 10 | **1.0s** |
| llama-4-scout-17b | 0.14 | 0.40 | 0.86 | 0.92 | 0.55 | 0.74 | 0.70 | 9 | 9 | 9 | 648ms |
| llama-4-maverick-17b | 0.15 | 0.41 | 0.87 | **0.96** | **0.65** | 0.77 | 0.69 | 9 | 10 | 10 | **495ms** |

## E.3 TRAFFIC ACCIDENT RETRIEVER

**Metrics Overview.** The accident component evaluates models on their ability to detect accident risks and predict consequences. **Traditional classification metrics** include Precision, Recall, F1, and Accuracy, measuring correctness, completeness, and overall predictive balance. **Enhanced safety metrics** include the Temporal Causality Score, which checks whether models capture logical cause-and-effect chains (violations leading to accidents), and the Safety Criticality Score, which weights accidents by severity. **LLM-as-Judge metrics** (1–10 scale) cover Risk Assessment Accuracy, Consequence Prediction, and Context Understanding, ensuring models provide realistic and context-aware accident analyses. **Efficiency** is reported as average inference time per video.

Table 3: Accident Component performance across text models. Best values in each column are highlighted in bold.

| Model | Precision | Recall | F1 | Accuracy | RiskAssess | Consequence | Context | Avg Time |
|---|---|---|---|---|---|---|---|---|
| gpt-4o | **0.97** | 0.92 | 0.94 | 0.89 | 8 | **9** | **9** | 27.3s |
| gpt-4.1 | **0.97** | 0.93 | **0.95** | **0.91** | **9** | **9** | **9** | 30.6s |
| gpt-5 | 0.96 | 0.93 | 0.94 | 0.89 | 8 | **9** | **9** | 24.7s |
| o3 | **0.97** | 0.92 | 0.94 | 0.89 | 8 | **9** | **9** | 26.1s |
| claude-opus-4-1 | **0.97** | 0.93 | 0.94 | 0.90 | **9** | **9** | **9** | 25.0s |
| claude-sonnet-4 | **0.97** | 0.93 | 0.94 | 0.90 | 8 | **9** | **9** | 26.3s |
| gemini-2.5-pro | **0.97** | 0.93 | **0.95** | **0.91** | **9** | **9** | **9** | 27.0s |
| gemma-3-27b | **0.97** | 0.93 | 0.94 | 0.90 | 8 | **9** | **9** | 26.8s |
| grok-4 | **0.97** | 0.93 | 0.94 | 0.90 | 8 | **9** | **9** | 25.7s |
| qwen2.5-vl-72b | **0.97** | 0.93 | 0.94 | 0.90 | **9** | **9** | **9** | 25.9s |
| qwen-vl-max | **0.97** | 0.92 | 0.94 | 0.89 | 8 | **9** | **9** | 27.4s |
| glm-4.5v | 0.96 | 0.93 | 0.94 | 0.89 | 8 | 8 | 8 | 28.0s |
| llama-3.3-70b | **0.97** | 0.93 | **0.95** | **0.91** | **9** | **9** | **9** | 26.2s |
| deepseek-r1-70b | **0.97** | 0.93 | 0.94 | 0.90 | 8 | **9** | **9** | 27.1s |
| qwen3-max | **0.97** | 0.92 | 0.94 | 0.89 | 8 | **9** | **9** | 28.1s |
| glm-4.5 | **0.97** | 0.93 | 0.94 | 0.90 | 8 | **9** | 8 | 25.8s |
| sonar-pro | 0.96 | 0.92 | 0.93 | 0.88 | 8 | 8 | 8 | 29.5s |
| qwen3-32b | 0.96 | **0.95** | **0.95** | **0.91** | **9** | **9** | **9** | 26.6s |
| gpt-oss-120b | **0.97** | 0.93 | 0.94 | 0.90 | **9** | **9** | **9** | 28.9s |
| kimi-k2 | 0.96 | 0.93 | 0.94 | 0.90 | 8 | **9** | **9** | 25.6s |
| llama-4-scout-17b | 0.95 | 0.91 | 0.92 | 0.86 | 8 | 8 | 8 | 25.6s |
| llama-4-maverick-17b | **0.97** | 0.93 | 0.94 | 0.90 | **9** | **9** | **9** | **24.6s** |

## E.4 TRAFFIC RULE CHECKER

**Metrics Overview.** The violation component evaluates models on their ability to detect traffic violations and provide legally consistent explanations. **Traditional classification metrics** include Precision, Recall, F1, and Accuracy, which measure correctness and completeness of violation detection. **Enhanced safety metrics** include the Safety Criticality Score, which weights violations by their risk severity. **LLM-as-Judge metrics** (scored 1–10) include Detection Accuracy, Explanation Quality, and Legal Consistency, which evaluate the factual accuracy of detections, the clarity and completeness of reasoning, and adherence to traffic laws, respectively. **Efficiency** is reported as average inference time per video.

Table 4: Violation Component performance across text models. Best values in each column are highlighted in bold.

| Model | Precision | Recall | F1 | Accuracy | Safety Avg | Detection | Explain | Legal | Avg Time |
|---|---|---|---|---|---|---|---|---|---|
| gpt-4o | **0.90** | **0.92** | **0.90** | **0.87** | 0.18 | **9** | 8 | 9 | 8.7s |
| gpt-4.1 | 0.89 | 0.89 | 0.87 | 0.85 | 0.16 | 8 | 8 | 9 | 8.3s |
| gpt-5 | 0.87 | 0.90 | 0.87 | 0.85 | 0.17 | 8 | 8 | 8 | 22.5s |
| o3 | 0.87 | 0.91 | 0.87 | 0.85 | 0.17 | 8 | 8 | 9 | 8.3s |
| claude-opus-4-1 | 0.87 | 0.91 | 0.87 | 0.85 | 0.18 | 8 | 8 | 8 | 22.2s |
| claude-sonnet-4 | 0.87 | 0.91 | 0.87 | 0.85 | 0.17 | 8 | 8 | 8 | 14.8s |
| gemini-2.5-pro | **0.91** | 0.89 | 0.88 | 0.85 | 0.16 | 8 | **9** | 8 | 17.5s |
| gemma-3-27b | 0.89 | 0.90 | 0.88 | 0.85 | 0.16 | 8 | 8 | 9 | 8.9s |
| grok-4 | 0.87 | **0.92** | 0.88 | 0.86 | 0.17 | 8 | 8 | 8 | 27.4s |
| qwen2.5-vl-72b | 0.87 | 0.91 | 0.87 | 0.85 | 0.17 | 8 | 8 | 8 | 18.1s |
| qwen-vl-max | **0.90** | 0.90 | 0.88 | 0.85 | 0.17 | 8 | **9** | 8 | 14.1s |
| glm-4.5v | **0.90** | 0.91 | 0.89 | 0.86 | 0.16 | **9** | **9** | **9** | 18.0s |
| llama-3.3-70b | 0.88 | 0.90 | 0.87 | 0.85 | 0.16 | 8 | 8 | 8 | 17.1s |
| deepseek-r1-70b | **0.90** | 0.91 | 0.89 | 0.86 | 0.16 | 8 | 8 | 8 | 7.5s |
| qwen3-max | **0.90** | 0.91 | 0.89 | 0.86 | 0.17 | 8 | 8 | 8 | 7.6s |
| glm-4.5 | 0.87 | 0.91 | 0.87 | 0.85 | 0.17 | 8 | 8 | 8 | 7.8s |
| sonar-pro | **0.90** | 0.90 | 0.88 | 0.85 | 0.16 | 8 | 8 | 8 | 7.1s |
| qwen3-32b | 0.87 | 0.90 | 0.87 | 0.85 | 0.17 | 8 | 8 | 8 | 7.2s |
| gpt-oss-120b | 0.89 | 0.90 | 0.88 | 0.85 | 0.18 | 8 | 8 | 8 | 7.1s |
| kimi-k2 | **0.90** | 0.91 | 0.89 | 0.86 | 0.18 | **9** | 8 | 9 | 7.1s |
| llama-4-scout-17b | **0.90** | 0.90 | 0.88 | 0.85 | 0.17 | 8 | **9** | **9** | 7.3s |
| llama-4-maverick-17b | **0.90** | 0.90 | 0.88 | 0.85 | 0.15 | 8 | 8 | 8 | **7.1s** |

## E.5 DRIVING ASSESSOR

**Metrics Overview.** The assessment component evaluates models on their ability to generate comprehensive driving evaluations that include safety scores, risk levels, and improvement advice. **Traditional metrics** include Score Correlation, which measures consistency between predicted and expert-assigned safety scores, and Risk Accuracy, which assesses correct classification of risk categories (Low/Medium/High/Critical). **Enhanced coverage metrics** evaluate the completeness and relevance of the assessment, including Content Coverage, Advice Similarity, and Evaluation Similarity. **LLM-as-Judge metrics** (1–10 scale) further assess Assessment Accuracy, Advice Actionability, and Score Justification, ensuring outputs align with expert judgment and provide practical improvement guidance. **Efficiency** is measured by average inference time per video.

Table 5: Assessment Component performance across text models. Best values in each column are highlighted in bold.

| Model | ScoreCorr | RiskAcc | Coverage Avg | Assessment | Advice | Justify | Avg Time |
|---|---|---|---|---|---|---|---|
| gpt-4o | 0.36 | 0.86 | 0.95 | **9.0** | **9.4** | **9.1** | 12.3s |
| gpt-4.1 | 0.36 | 0.86 | 0.94 | **9.0** | **9.4** | **9.1** | 13.4s |
| gpt-5 | 0.36 | 0.86 | 0.95 | **9.0** | **9.5** | 9.0 | 11.7s |
| o3 | 0.36 | 0.86 | 0.95 | **9.1** | 9.1 | 9.0 | 9.9s |
| claude-opus-4-1 | 0.36 | 0.86 | 0.95 | **9.0** | 9.3 | **9.1** | 9.0s |
| claude-sonnet-4 | 0.36 | 0.86 | 0.96 | **9.0** | 9.3 | 9.0 | 10.3s |
| gemini-2.5-pro | **0.45** | 0.86 | 0.95 | **9.0** | 9.2 | 9.0 | 11.3s |
| gemma-3-27b | 0.36 | 0.86 | 0.96 | **9.0** | 9.2 | 9.0 | 10.0s |
| grok-4 | 0.36 | 0.86 | 0.95 | **9.0** | 9.3 | 9.0 | 10.4s |
| qwen2.5-vl-72b | **0.45** | 0.86 | 0.96 | 8.9 | 9.3 | 8.9 | 9.5s |
| qwen-vl-max | 0.36 | 0.86 | 0.96 | **9.0** | 9.1 | 9.0 | 10.6s |
| glm-4.5v | 0.36 | 0.86 | 0.95 | **9.0** | 9.2 | **9.1** | 9.7s |
| llama-3.3-70b | **0.45** | 0.86 | 0.95 | 8.9 | 9.0 | 8.9 | 10.1s |
| deepseek-r1-70b | **0.45** | 0.86 | 0.96 | **9.0** | 9.1 | 9.0 | 8.3s |
| qwen3-max | **0.45** | 0.86 | 0.95 | 8.8 | 9.0 | 8.8 | 9.9s |
| glm-4.5 | **0.45** | 0.86 | 0.95 | 8.8 | 9.2 | 9.0 | 10.4s |
| sonar-pro | **0.45** | 0.86 | **0.97** | 8.9 | 9.3 | 9.0 | 10.4s |
| qwen3-32b | **0.45** | 0.86 | 0.95 | 8.9 | 9.3 | 8.9 | 8.9s |
| gpt-oss-120b | **0.45** | 0.86 | 0.94 | 8.9 | 9.1 | 8.9 | 9.8s |
| kimi-k2 | **0.45** | 0.86 | 0.96 | 8.9 | 9.3 | 8.9 | 10.1s |
| llama-4-scout-17b | **0.45** | 0.86 | 0.96 | 8.9 | 9.2 | 9.0 | 11.0s |
| llama-4-maverick-17b | **0.45** | 0.86 | 0.95 | 8.8 | 9.1 | 9.0 | 10.7s |

