# OpenReview forum: "Context-Aware Multi-Agent Safety Evaluation for Autonomous Driving"
_ICLR.cc/2026/Conference — Submitted to ICLR 2026_

### Official Review · Reviewer_DZoZ · 2025-10-29

**Soundness:** 2
**Presentation:** 3
**Contribution:** 2
**Rating:** 4
**Confidence:** 3

**Summary:**

This paper proposes DriveEval, a context-aware multi-agent framework designed for the safety evaluation of autonomous driving systems. It leverages VLM, LLM and external  knowledge bases  to address  insufficient coverage of unforeseen long-tailed scenarios, poor interpretability of unsafe behaviors, and limited adaptability across regions.  DriveEval consists of five specialized agents: Data Annotator, Scene Extractor, Rule Checker, Accident Retriever, and Driving Assessor, each responsible for modular subtasks in the evaluation pipeline. Evaluations are conducted on a custom dataset (200 dashcam clips with human-annotated safety reports) and two knowledge base datasets (derived from the Pennsylvania Driver's Manual and Nexar accident videos). Results show that DriveEval's assessments align closely with human annotations, enabling effective identification of safety risks and providing actionable improvement recommendations.

**Strengths:**

The framework decomposes the complex AD safety evaluation task into specialized, optimized agents. This design enhances evaluation precision through task specialization, and supports modular expansion.

By leveraging RAG to link evaluations with traffic rules and historical accidents. DriveEval addresses the black-box limitation of traditional methods. It generates structured safety reports that include quantitative scores, qualitative narrative evaluations, lists of strengths/weaknesses, and actionable advice.

**Weaknesses:**

Time efficiency may be a problem, which limits the suitability for real-time AD evaluation, a critical requirement for on-board or low-latency safety assessment scenarios.

The framework is evaluated on a relatively small dataset  and lacks large-scale testing, comparison with baselines, and ablation studies. Overall, the experiment is rather weak, making it difficult to judge its technical effectiveness.

**Questions:**

As observed , optimal models differ across agents (e.g., gpt-4o for the Data Annotator, llama-3.3-70b for the Scene Extractor), introducing challenges in integration and potential operational overhead for real-world deployment.  Can you provide guidance on simplifying model selection and reducing integration complexity?

When outputs from upstream agents conflict (e.g., the Rule Checker identifies no traffic violation, but the Accident Retriever finds analogous historical accident risks), how does DriveEval resolve these inconsistencies? Does the Driving Assessor include a dedicated mechanism to synthesize contradictory inputs into a coherent assessment?

You mentioned that annotated accident data is leveraged with RAG. Can new types of accidents (not present in the initial Nexar dataset) can be incorporated over time?

---

### Official Review · Reviewer_kMee · 2025-11-01

**Soundness:** 2
**Presentation:** 3
**Contribution:** 2
**Rating:** 2
**Confidence:** 3

**Summary:**

The paper proposes DriveEval, a framework for evaluating the safety of Autonomous Driving (AD) systems from sensor data (like dashcam videos). The primary contribution is a context-aware and interpretable evaluation paradigm that aims to augment traditional opaque metrics.

**Strengths:**

The paper tackles a problem of ensuring the safety of AD systems, which is significant for current intelligent transportation systems. The paper also presents a well-engineered solution towards the comprehensive assessment. The paper is clearly written, and the proposed architecture is well-explained. The problem is strongly motivated in the introduction .

**Weaknesses:**

This paper, in its current form, appears to be significantly below the bar for publication due to a lack of technical novelty, an over-engineered design, and significant absence of empirical results. The framework is a systems-integration paper that combines existing, off-the-shelf components (VLMs, LLMs, RAG, vector/graph databases)  without proposing any new model, algorithm, or novel methodology. It's a well-engineered architecture but lacks of significant methodological novelty. The quantitative results (which are put in the appendix section) lack of significant analysis and there's no enough qualitative examples to illustrate the difference behind the metrics, which are really hard to follow and thus diminish the readability of the paper.

**Questions:**

See weaknesses.

---

### Official Review · Reviewer_V9kb · 2025-11-01

**Soundness:** 2
**Presentation:** 3
**Contribution:** 2
**Rating:** 2
**Confidence:** 5

**Summary:**

This paper presents a context-aware multi-agent framework for autonomous driving safety evaluation, where multiple LLM/VLM agents analyze sensor data to detect safety risks and suggest improvements.
While the idea is interesting, the contribution is unclear and the framework appears closer to dataset annotation and classification than to genuine safety evaluation. The methodology lacks sufficient detail on prompting, RAG implementation, and sensor data processing. Comparisons with relevant state-of-the-art works are also missing.
Overall, the work is not yet mature enough for publication and would benefit from clearer methodological explanations, stronger experimental validation, and broader literature comparison.

**Strengths:**

•	The paper is well-structured and relatively easy to follow.
•	The appendix provides the prompting templates and model performance comparisons, which help with reproducibility.

**Weaknesses:**

1.Unclear Motivation and Contribution:
The motivation and claimed contributions are not well aligned with autonomous driving safety evaluation. The proposed framework functions more as a classification and annotation pipeline for driving scenarios rather than a systematic approach to safety assessment.
2.Incomplete Related Work:
The discussion of related work is limited. Several recent studies using LLMs and VLMs for scenario understanding, risk analysis, and autonomous driving evaluation should be included to establish a stronger connection to the state of the art.
3.Limited Practical Relevance:
The framework does not demonstrate applicability to real-world or closed-loop safety evaluation. Therefore, the claim of “recommending actionable improvements” appears overstated without validation in realistic deployment settings.
4.Methodological Ambiguity:
Key methodological details are missing. It is unclear how the RAG process is implemented (e.g., embedding models, retrieval strategies) or how structured sensor data from the ego vehicle and surrounding agents are represented for LLM/VLM interpretation.
5.Experimental Insufficiency:
The experiments rely on template-based prompting and few-shot examples but lack ablation studies and comparisons with open-source baseline models (e.g., 3B–7B) which is most used for the real world deployment. The evaluation does not convincingly support the claimed performance or real-world applicability.
6.Weak Result Interpretation:
Although each of the five agents achieves around 90% accuracy individually which is very hard, the framework operates sequentially, meaning that errors accumulate across stages. As a result, the overall end-to-end accuracy drops below 60%. The paper should analyze this performance degradation, explain the error propagation between agents, and discuss how model scalability, token efficiency, and reasoning depth affect the overall system reliability.

**Questions:**

1.  How does the proposed framework advance safety evaluation, rather than functioning mainly as a dataset annotation or classification pipeline?
2.  How can the system be integrated into real-world autonomous driving evaluation or validation workflows?
3.  How are sensor data from the ego vehicle and other agents represented for LLM/VLM processing, especially under token-length constraints?
4.  Could the authors clarify the RAG implementation details, including embedding, retrieval, and integration mechanisms?
5.  Have the authors performed ablation studies on prompting strategies and few-shot examples, and how does performance scale across 3B–7B open-source models suitable for real-world deployment?
6.  What metrics and baselines were used, and how do they correspond to human or real-world safety assessments?

---

### Official Review · Reviewer_7mUE · 2025-11-01

**Soundness:** 3
**Presentation:** 3
**Contribution:** 2
**Rating:** 4
**Confidence:** 3

**Summary:**

This paper introduces DriveEval, a context-aware multi-agent framework for autonomous driving safety evaluation. The framework processes raw sensor data through a pipeline of specialized agents: a Data Annotator , a Traffic Scene Extractor, a Traffic Rule Checker, a Traffic Accident Retriever, and a Driving Assessor. These agents leverage external knowledge bases of traffic rules and historical accidents to generate structured, interpretable safety reports. The core contribution is a modular system designed to provide explainable assessments. Experiments evaluate various LLMs/VLMs within each agent role using a custom dataset of 200 dashcam clips, reporting performance across multiple metrics for each component.

**Strengths:**

The paper presents a well-structured multi-agent framework for autonomous driving safety evaluation, leveraging modern LLMs and VLMs. Its strengths include a comprehensive evaluation of numerous models across different agent roles and the creation of a novel annotated dataset. However, the technical novelty of the core agent designs is limited, as they primarily rely on standard prompting and RAG techniques. The practical utility and scalability in real-world deployment scenarios are not sufficiently demonstrated , and the experimental results, while extensive, lack statistical validation and comparison to non-LLM baselines.

**Weaknesses:**

1.Limited Technical Novelty in Core Agents: The designs of the key analytical agents (Scene Extractor, Rule Checker, Accident Retriever, Driving Assessor) are primarily based on standard LLM prompting and existing paradigms like RAG and GraphRAG, without introducing significant algorithmic innovations beyond the modular composition.

2. Lack of Baseline Comparisons: The experimental evaluation focuses solely on comparing different LLMs/VLMs within the proposed framework. It does not include comparisons against established non-LLM-based safety evaluation methods or ablations of the framework itself against simpler monolithic evaluators, making it difficult to assess the added value of the multi-agent, context-engineered approach.

3. Absence of Statistical Significance and Robustness Analysis: The performance results presented in Tables 1-5 are point estimates. There is no reporting of standard deviations, confidence intervals, or results from multiple runs, which is crucial for assessing the statistical significance and robustness of the findings, especially given the inherent variability of LLM outputs.

4. Unclear Real-World Applicability and Scalability: The paper lacks a discussion on the computational cost, latency, and practical deployment challenges of the full pipeline for real-time or large-scale safety evaluation. The use of multiple powerful LLMs/VLMs in sequence suggests high operational costs, but this is not quantified or addressed .

5. Insufficient Analysis of Agent Failures and Error Propagation: The multi-agent design is susceptible to error propagation (e.g., inaccuracies from the Data Annotator affecting downstream agents). The paper does not analyze such failure modes or the overall robustness of the pipeline to errors in individual components.

**Questions:**

1. Include comparisons with state-of-the-art non-LLM safety evaluation metrics or systems on the DriveEval dataset to better position the framework's performance advantages.

2. Report the mean and standard deviation of key metrics over multiple independent runs (with different random seeds if applicable) for a subset of models to demonstrate the statistical robustness of the results.

3.Add a discussion or experiment addressing the computational efficiency and latency of the entire DriveEval pipeline, and explore potential optimizations for practical use cases.

4.Perform a qualitative error analysis, presenting case studies where the framework succeeded or failed, and analyze the root causes of failures within the agent pipeline.

5.Clarify the annotation process and inter-annotator agreement for the creation of the ground-truth safety reports in the DriveEval dataset to establish its reliability.

---

### Meta-Review · Area_Chair_H3WY · 2026-01-07

**Summary:**

The authors did not submit the rebuttal. All four reviewers were unconvinced on the positive side; they agreed that this work requires additional effort to meet the acceptance bar of ICLR. Thus, I am inclined not to accept this draft at this stage. Thank you for your effort. It is an interesting work. I hope the input from the reviewers will help you further improve this work.

**Reviewer Concerns:**

This work has limited novelty and experimental results.

**Reviewer Scores:**

The reviewers' concerns reflect the limitations in novelty and experiments of this work.

---

### Decision · Program_Chairs · 2026-01-26

Reject